# The Role of Cancer Stem Cell Markers in Ovarian Cancer

**DOI:** 10.3390/cancers16010040

**Published:** 2023-12-20

**Authors:** Karolina Frąszczak, Bartłomiej Barczyński

**Affiliations:** 1st Chair and Department of Oncological Gynaecology and Gynaecology, Medical University in Lublin, 20-081 Lublin, Poland; karolina.fraszczak@umlub.pl

**Keywords:** cancer stem cells, ovarian cancer, markers, CD44, CD133, ALDH1, CD117, CD24

## Abstract

**Simple Summary:**

This manuscript focuses on cancer stem cells and the diagnostic potential of selected biomarkers of these cells in ovarian cancers. Ovarian cancer has nonspecific clinical symptoms. Also, there are no reliable diagnostic and prognostic biomarkers. For these reasons, nearly 70% of patients are diagnosed with ovarian cancer at the metastatic stage. The role of CSCs in ovarian cancer initiation and progression and their impact on resistance to therapy are still the subjects of many studies; however, the results are sometimes conflicting due to the heterogeneity and plasticity of CSCs. The identification of CSC phenotypes could contribute to the development of more effective diagnostic and therapeutic strategies in ovarian cancer. This is especially important since strategies to overcome resistance to conventional treatments and prolong patient survival are highly awaited.

**Abstract:**

Ovarian cancer is the most lethal gynaecological cancer and the eighth most common female cancer. The early diagnosis of ovarian cancer remains a clinical problem despite the significant development of technology. Nearly 70% of patients with ovarian cancer are diagnosed with stages III–IV metastatic disease. Reliable diagnostic and prognostic biomarkers are currently lacking. Ovarian cancer recurrence and resistance to chemotherapy pose vital problems and translate into poor outcomes. Cancer stem cells appear to be responsible for tumour recurrence resulting from chemotherapeutic resistance. These cells are also crucial for tumour initiation due to the ability to self-renew, differentiate, avoid immune destruction, and promote inflammation and angiogenesis. Studies have confirmed an association between CSC occurrence and resistance to chemotherapy, subsequent metastases, and cancer relapses. Therefore, the elimination of CSCs appears important for overcoming drug resistance and improving prognoses. This review focuses on the expression of selected ovarian CSC markers, including CD133, CD44, CD24, CD117, and aldehyde dehydrogenase 1, which show potential prognostic significance. Some markers expressed on the surface of CSCs correlate with clinical features and can be used for the diagnosis and prognosis of ovarian cancer. However, due to the heterogeneity and plasticity of CSCs, the determination of specific CSC phenotypes is difficult.

## 1. Introduction

Ovarian cancer is the most lethal gynaecological cancer and the eighth most common female cancer [1,2]. An estimated 295,000 women are diagnosed with ovarian cancer every year worldwide, and more than 184,000 women per year die due to the disease [3]. Due to nonspecific clinical manifestations, ovarian cancer is known as a silent killer [4]. Despite the significant development of technology, the early diagnosis of ovarian cancer remains a clinical problem. Ovarian cancer is usually diagnosed in women aged 55–64 years [4]. In nearly 70% of patients, ovarian cancer remains undetected due to the absence of disease symptoms until stages III–IV, when it spreads locally [4]. The late diagnosis of ovarian cancer contributes to reduced survival [5]. Recently, cancer stem cells (CSCs) have attracted the interest of researchers. CSCs are a subpopulation of tumour cells that are crucial for tumour initiation and are responsible for tumour recurrence resulting from chemotherapeutic resistance [6]. The presence of these cells is identifiable based on specific surface molecules (including CD44, CD133, aldehyde dehydrogenase 1 (ALDH1), CD117, and CD24) or intracellular enzyme activity [7]. Targeting CSCs could offer a promising approach that could increase the effectiveness of ovarian cancer treatment.

## 2. Ovarian Cancer

The ovary comprises various cell types; therefore, cancers in this organ are heterogeneous in terms of aetiology, phenotypes, molecular factors, progression, metastasis, response to chemotherapy, and prognosis [8,9]. Most ovarian tumours develop from epithelial cells (90% of malignant tumours originate from these cells), sex cord–stromal cells (5–6% of malignant cancers), and germ cells (<3% of malignant tumours) [10]. Epithelial ovarian cancers may originate from surface epithelial cells or intra-epithelial carcinomas, and they can be either low-grade serous carcinomas (genetically sustained) or invasive high-grade serous carcinomas (genetically unstable) [6,11]. Ovarian tumours originating from epithelial cells can be divided into five major histological subtypes: high-grade serous carcinoma (70% of all ovarian carcinomas), low-grade serous carcinoma (5%), endometrioid carcinoma (10%), clear cell carcinoma (6–10%), and mucinous carcinoma (3–4%) [12]. High-grade serous ovarian carcinoma has recently been suggested to originate from the fallopian tube and ovarian surface epithelium [12]. It is a fast-growing, very aggressive neoplasm within the mesentery and omentum, and it is usually accompanied by ascites [13]. Low-grade serous carcinoma originates from the fallopian tube; clear cell carcinoma, endometrioid carcinoma, and seromucinous carcinoma develop from endometriosis. The source of a malignant Brenner tumour is the transitional epithelium, while mucinous carcinoma arises from germ cells [13].

Based on molecular characteristics, epithelial ovarian cancers can be subdivided into type I cancers (endometrioid, clear cell, mucinous, and low-grade serous carcinomas), which are rarely associated with a *TP53* mutation, usually indolent, and present at a younger age, and type II carcinomas (high-grade serous carcinoma, undifferentiated carcinoma, and carcinosarcoma), which have unknown precursor lesions, frequently carry a *TP53* mutation, and are aggressive in nature [13]. Type I tumours are usually slow-growing, low-grade, genetically more stable, and restricted mostly to the ovary at diagnosis. Type II tumours are highly proliferative, high-grade, and genetically unstable tumours that spread to the peritoneum or the omentum [14]. Patients with type II ovarian tumours are also frequently carriers of mutations in *NOTCH*, *RB1*, and *FOXM1* [13].

Sex cord–stromal ovarian cancers are thought to originate from stromal and primitive ovarian sex cord cells; they can be composed of a singular cancerous cell type or a mixture of various cell types, and their cells can produce androgens, corticoids, and oestrogens [15,16]. Germ cell ovarian cancers (including dysgerminomas, embryonal carcinomas, and yolk sac tumours) emerge in primitive cells of the embryonic gonad and form a diverse group of benign and malignant tumours harbouring abnormalities in chromosome 12p [17].

Ovarian cancer risk factors include a family history of the disease; mutations in *BRCA1/BRCA2* genes; early menarche; being nulliparous; the presence of endometriosis; a history of breast, uterine, or colorectal cancer; and late menopause [18]. The presence of *BRCA1/2* mutations is associated with the invasive high-grade serous histotype, as well as with an improved 5-year survival rate due to an improved response to platinum-based chemotherapy [19]. Apart from *BRCA1/2*, mutations in *TP53*, *BARD1*, *C14EK2*, *PALB2*, *MRE11*, *KRAS*, *BRAF*, *PTEN*, CTNNB1 (encoding β-catenin), *BRIP1*, *NBN*, *MSH6*, *RAD50*, and *RAD51C* also contribute to the onset of ovarian cancer [3,20]. Various ovarian cancer cells overexpress high-mobility-group AT-hook (HMGA) proteins, which play a key role in the regulation of cell growth, differentiation, neoplastic transformation, and apoptosis [21]. Somatic mitochondrial DNA may also be involved in the development and progression of ovarian cancers.

Ovarian cancer therapy is adjusted to disease stage. A common treatment for any ovarian cancer stage includes complete cytoreductive surgery (the resection of all macroscopically visible lesions) and chemotherapy [22]. At stages I–IIA, surgical removal of the macroscopic tumour with a biopsy of the lymph nodes and other sites within the abdominal cavity is of high importance [1]. At more advanced stages, the goal is either the complete debulking of the visible tumour or neo-adjuvant chemotherapy and subsequent resection of the cancer [23,24]. Standard chemotherapy involves the combination of carboplatin with doxorubicin, paclitaxel, or docetaxel [25,26]. Additional chemotherapy with carboplatin or cisplatin and paclitaxel is necessary for patients at almost all tumour stages (except for International Federation of Gynaecology and Obstetrics (FIGO) stage Ia). The intraperitoneal administration or dose-dense schedules of cisplatin and paclitaxel have been found to increase overall survival [27,28]. Improved progression-free survival has been demonstrated after the additional administration of angiogenesis inhibitors, as well as maintenance therapy with poly ADP ribose polymerase (PARP) inhibitors (in platinum-sensitive disease) [29]. Other chemotherapeutic agents, including bevacizumab, gemcitabine, niraparib, olaparib, topotecan, and trabectedin, have been evaluated in clinical trials in patients with platinum-resistant disease, but they have not shown significant therapeutic success, as evidenced by a lack of considerable improvement in the progression-free interval [30]. Despite a good initial response to treatment, the relapse rate within 2 years is as high as 70–80% in patients with advanced-stage disease and 20% in those with early-stage disease [14,31]. The success of these therapies is limited due to chemotherapy-related toxicity, the development of drug resistance, and cancer recurrence [32]. Chemoresistance is an important obstacle in the treatment of ovarian cancer and is associated with the progression of tumorigenesis [33]. Chemoresistance may be related to the intratumoural heterogeneity of CSCs [3]. Given the role of CSCs in tumour development and progression, they should be considered highly important therapeutic targets. 

## 3. Cancer Stem Cells

CSCs are a specialised group of cells constituting a small percentage of the tumour mass; they display pluripotency, anchorage-independent growth, self-renewal properties, and the ability to trigger tumorigenesis (higher tumour-initiating potential) [34,35]. CSCs were initially described in haematological cancers; however, they were later identified in solid tumours [36]. The first ovarian CSCs were isolated and cultured by Bapat et al. [37]. Both the isolated single tumorigenic clone and the clone that evolved from the first one showed differentiation features. Moreover, they became the source of new tumours after transplantation into the mouse peritoneal cavity. Based on these findings, stem cell transformation was suggested to be the cause of ovarian cancer, as well as the higher aggressiveness of this disease [37]. The presence of CSCs in ovarian cancers may also underlie the heterogeneity of this disease. Understanding the regulation of the relative balance between self-renewal and differentiation is essential to comprehend cancer cell proliferation. The self-renewal of stem cells is crucial for their lifelong persistence [30]. Both the origin and mechanisms underlying the activation of CSCs that are capable of reproducing histological tumour characteristics remain not well understood [38].

There are several theories concerning the origin of CSCs since their precursors have not been unequivocally identified due to the fact that they may differ between various cancers. The model of clonal evolution suggests that all cancer cells are biologically uniform and have the same potential to grow and develop. Other theories proposed that CSCs can be formed from somatic stem cells, normal progenitor cells, or normal differentiated cells that underwent a mutation changing their phenotype and providing crucial cellular properties [39]. Horizontal gene transfer, genomic instability, and microenvironmental changes (including hypoxia, stress, ionising radiation, and wounding) have been suggested to contribute to the transformation of cells into CSCs [40,41]. Stem cell differentiation takes place in a specialised niche (the microenvironment) that stimulates self-renewal via cell–cell communication or the release of paracrine factors [42]. TGF-β1-mediated EMT has been suggested to be involved in the acquisition of a CSC-like phenotype [43]. The formation of CSCs could be initiated by cell fusion and the subsequent metabolic reprogramming of non-CSCs [40,41]. The thesis that CSCs originate from normal stem cells is supported by the similarities between normal stem cell and CSC features, including self-renewal via asymmetric division; regulation by Wnt, Hedgehog, or Notch pathways; and the ability to differentiate into multiple progenitor cell types [44,45]. However, these two types of cells differ in the control and regulation of self-renewal, which is lost in CSCs due to genetic mutations and epigenetic changes. CSCs show unlimited proliferative, survival potential and much higher plasticity than normal stem cells [44]. The ability of CSCs to form tissues and organs enables the formation of tumour tissues [42]. Finally, high telomerase activity in CSCs translates into a prolonged life span [42]. CSCs are characterised by specific metabolic features, including increased glycolytic functions compared with differentiated tumour cells [46]. Ovarian CSCs expressing CD44 and CD117 were shown to have high levels of mitochondrial reactive oxygen species (ROS). Therefore, it was suggested that, under nutrient deprivation and stress conditions, mitochondrial electron respiratory chain functioning is directed towards cell preservation [47]. 

The acquisition of oncogenic genetic and epigenetic alterations determines aggressiveness, invasiveness, and treatment resistance [48,49]. Cancer cells may gain stemness due to an oncogenic hit [50]. The hierarchical model proposes that a specific cell with tumorigenic potential becomes the first abnormal CSC after it escapes regulation [51]. The cellular plasticity model combines these two models and proposes that CSCs are capable of switching to differentiated states [44]. The potential to reverse differentiation can either be inherited or gained through mutations. Differentiated tumour cells may switch back to stem cells as a result of stimuli within the tumour microenvironment and/or intrinsic processes [51]. CSCs are characterised by the expression of undifferentiated stem cell markers, including *ABCG2*, *BMI1*, *NANOG*, *NESTIN*, and *OCT4* [52]. Moreover, they can differentiate into ovarian marker-expressing cells. CSCs are capable of lifelong regeneration and differentiation in an asymmetric mode [1]. Asymmetric divisions result in the formation of one daughter stem cell and one differentiated cell, and this feature enables the initiation of the neoplastic process [53]. CSCs can also improve cancer survival via EMT [54]. 

Apart from CSCs, the cancer microenvironment comprises immune cells (T cells, natural killer cells, macrophages, etc.), cancer-associated fibroblasts, the extracellular matrix, endothelial cells, mesenchymal stem cells, and secreted cytokines and growth factors [42]. Such a microenvironment appears to provide optimal conditions for the differentiation, proliferation, self-renewal, and development of a heterogeneous cancer cell population [55]. Cells surrounding the cancer niche also secrete factors that stimulate CSC plasticity and survival and that physically protect CSCs from chemotherapeutic agents [56]. However, CSCs have been found to produce factors that stimulate the recruitment and activation of niche components.

Recent studies have demonstrated that CSCs are involved in cancer progression, metastasis, and recurrence, as well as resistance to radiotherapy and chemotherapy [57,58]. Stem cells are also characterised by dormancy (quiescence) that can be transient or long term and is associated with CSC resistance to chemotherapy [59]. Moreover, quiescence is the reason for local recurrence and/or distant metastasis after a long duration of remission. In solid tumours, two subpopulations of stem cells have been identified: resident cancer stem cells and migrating stem cells. Resident cancer stem cells are involved in the initiation of the disease, while migrating stem cells contribute to cancer propagation and metastasis [60]. Based on this finding, it has been suggested that heterogeneous clones forming tumour stem cells from various mutations have different roles in tumour development [61]. Populations of CSCs displaying specific metabolic profiles, phenotypes, and clonogenic potential have been observed to metastasise to particular organs [62]. 

Several studies have provided evidence for the link between the occurrence of CSCs and resistance to chemotherapy, subsequent metastases, and cancer relapses [55,63,64]. So far, three drug resistance mechanisms, namely, ATP-binding cassette (ABC) transporters, DNA repair signalling pathways, and aldehyde dehydrogenase (ALDH), have been described in CSCs [65]. Increased levels of various transporters, such as ABCA1, ABCB5, and ABCC3/MRP3, have been demonstrated in ovarian tumour tissues, and an increased expression of *ABCA1*, *ABCB1*/*MDR1*/*P-GP*, and *ABCG2*/*BCRP* has been demonstrated in ovarian CSCs [66,67,68]. Chemoresistance in CSCs is also associated with the B-cell lymphoma-2 (BCL-2) protein family, which affects apoptosis, survival, embryogenesis, haematopoiesis, and neurogenesis via the inhibition of the pro-apoptotic proteins BAX and BAK, as well as the release of cytochrome C [69]. Increased BCL-2 protein levels were found to be crucial for CSC survival and chemoresistance, whereas the downregulation of BCL-2 has been linked to increased sensitivity to chemotherapeutics, including oxaliplatin and 5-fluorouracil [65]. The Wnt/β-catenin and Notch signalling pathways are also involved in the development of chemoresistance in CSCs [70]. Notch signalling is primarily involved in angiogenesis, tumour progression, EMT, and self-renewal [71]. The third vital mechanism of CSC drug resistance involves the activity of ALDH. There are several isoforms of ALDH expressed primarily in the kidney and liver. The inhibition of ALDH has been shown to result in the sensitivity of CSCs to drugs [72]. More information on ALDH is presented in the section on CSC markers. 

Platinum resistance is an important problem in epithelial ovarian cancer. Different histotypes have been suggested to have different mechanisms of resistance [30]. Cells sensitive to platinum chemotherapy undergo apoptosis following treatment, while platinum-resistant cells persist, replenish, and contribute to early recurrence [73]. Several signalling pathways, such as Wnt, Sonic hedgehog (Shh), VEGF, TGF-β, Notch1, JAK-STAT, and PI3K/Akt/mTOR, have been suggested to be involved in CSC-mediated therapeutic resistance [74,75,76,77,78,79]. 

## 4. Interactions between CSC and Non-Tumour Cells

The initiation and progression of cancer have conventionally been perceived as cell-autonomous processes triggered by successive genetic and epigenetic modifications that drive cell transformation independently of external influences [80]. However, currently, it is increasingly recognised that tumours not only consist of tumour cells but also encompass diverse stromal cell types, such as fibroblasts, immune cells, adipose cells, and endothelial cells, which are present in the tumour microenvironment (TME). Cancer stem cells (CSCs), akin to normal stem cells, are regulated by intrinsic and extrinsic signals [81,82]. In the case of CSCs, these signals come from the TME. Recent studies have suggested that distinct subtypes of high-grade serous ovarian cancer, exhibiting differential responses to therapy, may be distinguished primarily by the composition of non-tumour cells in the surrounding TME [83,84].

In the ovarian cancer TME, interactions between CSCs and carcinoma-associated mesenchymal stem/stromal cells (CA-MSCs) are mediated by various secreted cytokines and growth factors [85]. Some researchers point to such paracrine interactions as factors implicated in the enrichment of CSC and their chemoprotection. A study utilising a model of ovarian malignant ascites incorporating both CSCs and CA-MSCs demonstrated that platelet-derived growth factor (PDGF) signalling in CSC-MSC heterospheroids significantly enhanced stemness, metastatic potential, and chemoresistance in CSCs [85]. The knockdown of platelet-derived growth factor subunit B (PDGFB) in mesenchymal stem/stromal cells abrogated these phenotypes in the heterospheroids. Moreover, Raghavan et al. [85] revealed an interaction between PDGF and Hedgehog signalling in ovarian cancer. These findings imply that blocking stromal signalling, particularly via PDGF-related pathways, accompanied by chemotherapy pressure renders tumour cells markedly more sensitive to chemotherapy. According to Raghavan et al. [85], the disruption of microenvironmental signals to tumour cells is of high importance to enhance response rates. They also implied that a combination of therapies targeting stromal signalling pathways, such as PDGF and Hedgehog, may provide an opportunity to counteract the tumorigenic, metastatic, and platinum-resistant phenotypes of ovarian CSCs.

Moreover, accumulating evidence suggests that immune cells not only contribute to cancer stem cell (CSC) expansion but also induce CSC-specific mechanisms for avoiding immune detection and destruction [80]. A bidirectional interaction between cancer cells and immune cells has been suggested. Specific immune cell types do not only drive CSC expansion, as recent findings have indicated that CSCs possess distinct capabilities to evade surveillance and immune-cell-mediated destruction [80,86]. Tumour-associated macrophages (TAMs) can be divided into tumour-suppressing M1-TAMs and tumour-promoting M2-TAMs, both infiltrating the tumour microenvironment and affecting tumorigenicity [87]. Raghavan et al. [88] observed that CSCs promoted the upregulation of the M2 macrophage marker CD206 within heterospheroids, which indicates an immunosuppressive program. Furthermore, preserved increased aldehyde dehydrogenase (ALDH) activity was observed in heterospheroids containing pre-polarised CD206+ M2 macrophages, suggesting a reciprocal interaction that triggers pro-tumoral activation and CSC self-renewal [88]. Also, increased levels of IL-10 and IL-6 cytokines were detected in CSC/M2 macrophage heterospheroids. CSC/M2 macrophage heterospheroids were found to exhibit reduced sensitivity to carboplatin and increased invasiveness. Moreover, Raghavan et al. [88] observed that CSC-derived WNT ligands induced the activation of CD206+ M2 macrophages, while macrophage-derived WNT ligands increased the abundance of ALDH+ cells within the CSC compartment of heterospheroids. Based on their findings, Raghavan et al. [88] suggested the significance of macrophage-initiated WNT signalling in maintaining stemness and driving chemoresistance and invasiveness. The paracrine activation of WNT in the course of interactions between CSC and M2 macrophages appears to form a positive feedback loop, contributing to a more aggressive phenotype. Therefore, targeting the WNT pathway emerges as a potential strategy to mitigate the CSC and M2 macrophage compartments within the tumour microenvironment [88]. 

Various clinical trials have confirmed that an increased abundance of tumour-associated macrophages (TAMs) is correlated with the poor survival of patients across various cancer types [89,90]. The tumour microenvironment secretes a variety of soluble factors that promote regular myeloid differentiation, transforming myeloid cells into immunosuppressive cells. This forms a tumour-promoting ‘macroenvironment’ that significantly hampers the effectiveness of cancer immunotherapy [91,92]. Myeloid-derived suppressor cells (MDSCs), a subset of immune cells within the tumour milieu characterised by elevated iNOS and arginase levels, exert a suppressive influence on T-cell activity [92]. These cells have been proposed to be heterogeneous and arise from myeloid cells under conditions of chronic inflammation, cancers, and infections [93]. The production of prostaglandin E2 (PGE2) by MDSCs appears to promote the stemness and expression of programmed death ligand 1 (PD-L1) in ovarian cancer stem cells expressing high levels of ALDH 1 through the activation of the PI3K/AKT/mTOR signalling pathway [94]. Based on the results of studies, an increased expression of PD-L1 is implicated in PD-L1-dependent T-cell suppression, thus fostering tumour growth and metastasis [95]. MDSCs are frequently accompanied by neutrophils in a chronic inflammatory state. Neutrophils stem from myeloid precursors and contribute to the innate immune response [80]. They can exhibit either antitumour N1 phenotypes or pro-tumour N2 phenotypes in cancer [96]. Tumour-associated neutrophils (TANs) were found to show functional plasticity across different malignancies [97]. Accumulating evidence implies a role of TANs in cancer metastasis. 

Also, the regulatory function of Treg cells, which are responsible for the suppression of autoimmunity and the modulation of immune function, has been found to decrease antitumour immunity in ovarian cancers, thus enabling tumour cells to evade any antitumour response [98]. The suppression of antitumor effects correlates with poor survival and cancer stemness. The presence of CSCs is accompanied by elevated levels of Treg cells, but Treg cell populations also increase along with CSC expansion during cancer progression, showing their tumour-promoting properties [99]. 

Cancer-associated fibroblasts (CAFs) are another element of the TME involved in a complex net of interactions. These cells are formed from resting fibroblasts via the activation mediated by the NF-κB and JAK-STAT pathways [80]. This activation is triggered by signalling molecules released by cancer cells or immune cells, including TGFβ, RTK ligands, IL1β, and IL6 [100]. CAFs, as key stromal components in the TME, engage in interactions with stromal and immune cells, thus influencing CSC plasticity and immune evasion. The reciprocal induction of TAM activity by CAFs is reported to play a pivotal role in promoting cancer stemness in soma cancers [101]. Activated CAFs acquire the capacity to produce extracellular matrix (ECM) components and essential molecules that sustain tumour growth and cancer stem cell (CSC) properties, thereby fostering therapeutic drug resistance [102]. Moreover, some studies demonstrated that factors secreted by CAFs induce epithelial–mesenchymal transition (EMT), further augmenting CSC properties [103].

Emerging evidence supports the existence of an obesity/cancer axis, indicating a positive correlation between adipose tissue and various cancers [80]. Multiple cancer types utilise fatty acids from surrounding adipose tissue by inducing adipocyte lipolysis, promoting energy metabolism (β-oxidation) and the biosynthesis of lipid-derived cell signalling molecules [104]. The adipocytes in adipose tissue contribute to sustaining CSC properties through paracrine secretion into the TME [105]. Cancer-associated adipocytes were found to exhibit distinct phenotypes and effects compared to normal adipocytes. Leptin activates tumour cell proliferation and migration. The adipocyte-associated secretion of IL-6 is involved in the Notch/Wnt/TGF-β signalling pathways, upregulating ALDH1A1, AXIN2, and LEF1 gene expression in the Wnt pathway [106]. This contributes to enhancing the invasiveness, metastasis, and angiogenesis of breast cancer. Hypoxia within the adipose tissue of obese individuals arises from inadequate blood perfusion due to a relatively low microvessel density [104]. This condition triggers angiogenesis while concurrently inhibiting macrophage migration and preadipocyte differentiation. Additionally, hypoxia augments fibrosis, suppresses immune cell recruitment, and imparts drug resistance capabilities to cancer cells [107]. The omentum is one of the primary sites for ovarian cancer metastasis. Omental adipocytes contribute to the early metastatic seeding process by secreting interleukin-6 (IL-6), IL-8, a tissue inhibitor of metalloproteinase 1 (TIMP1), and monocyte chemoattractant protein 1 (MCP1) [108]. In response, ovarian cancer cells activate Wnt/β-catenin signalling, inducing the dedifferentiation of omental adipocytes into mesenchymal stem-cell-like and myofibroblast-like fibroblasts expressing CD73, CD90, CD105, and alpha-smooth muscle actin (α-SMA). These fibroblasts further support ovarian cancer cell proliferation and migration [108]. Moreover, the IL-6 cytokine secreted by adipocytes plays a crucial role in the inhibition of mitochondria-triggered apoptosis in chemoresistant ovarian cancer stem cells (CSCs), particularly in the CD44+/MyD88+ cancer cell population [109]. IL-6 functions as a stimulator of various CSC types since it stimulates CSC properties via the induction of transcription factors such as OCT4, ZEB2, and NANOG. Moreover, IL-6 influences the expression of stemness and metastasis-related genes, including WNT5A, NODAL, WNT5B, SDF1, WNT7A, matrix metalloproteinase 2 (MMP-2), MMP-9 ZEB2, and TWIST [110]. 

Angiogenesis is a key process for sustaining tumour growth by ensuring an adequate supply of nutrients and oxygen [80]. Endothelial cells, particularly the vascular endothelial cells lining blood vessels, play a significant role in both maintaining cancer stem cells (CSCs) and facilitating tumour metastasis. The pivotal role of angiogenesis in tumour growth and metastasis is well established across various cancers. Nevertheless, the potential effects of angiogenesis-related genes (ARGs) in ovarian cancer (OC) necessitate further investigation. Ji et al. [111] revealed the substantial involvement of ARGs in the tumour–immune–stromal microenvironment, clinicopathological characteristics, and prognosis of OC patients. 

The observed relationship between CSCs and other cells within the tumour microenvironment (TME) suggests that developing novel therapeutics targeting CSC-TME interactions holds the potential to enhance clinical outcomes [2]. Main molecular mechanisms and signalling pathways related to CSCs are presented in Figure 1.

## 5. Targeting CSCs in Ovarian Cancer

Several completed and ongoing clinical trials have assessed the role of CSCs in ovarian cancer and undertook efforts to develop CSC-targeting strategies. Metformin has been found in previous studies of cell lines to destroy chemotherapy-resistant breast CSCs [112]. Also, in ovarian cancer cell lines, the use of metformin resulted in limited proliferation, reduced the percentage of ALDH+ CSCs, and hampered the sphere formation ability of ALDH+ cells in established cell lines and cells obtained from short-term patient tumour cell cultures [113]. Furthermore, metformin constrained the growth of ALDH+ CSC xenografts [113]. The NCT01579812 clinical trial assessed the impact of metformin as an anticancer agent [114]. The obtained results indicated a reduction in ALDH+ CD133+ CSCs and increased sensitivity to cisplatin ex vivo, which translated into a better-than-expected overall survival of patients receiving metformin, confirming the efficacy of the drug in various treatment combinations [115].

Ongoing trials mostly focus on the impact of CSC-directed chemotherapy on overall survival and progression-free survival (NCT03949283 and NCT03632798). The safety of and immune responses to a CSC vaccine have also been evaluated in a clinical trial involving women with metastatic ovarian adenocarcinoma; however, the results of this study have not yet been published. The preliminary results of animal studies demonstrated that CSC-primed antibodies and T cells could selectively target CSCs and present antitumor immunity [116]. Moreover, cytotoxic T lymphocytes obtained using peripheral blood mononuclear cells or splenocytes collected from CSC-vaccinated hosts were capable of killing CSCs in vitro. Data concerning completed and ongoing trials are summarised in Table 1. 

There are just a few clinical trials assessing the effectiveness and safety of CSC-targeted strategies. However, some potential anticancer substances have been studied in in vitro and in vivo studies. One of such substances is salinomycin, an antibiotic naturally isolated from *Streptomyces albus*, and it acts as an ionophore and stimulates cation transfer across biological membranes [117]. The results of preliminary studies have suggested that salinomycin selectively limits ovarian CSC growth and survival [117,118]. Lee et al. demonstrated that the combination of paclitaxel with salinomycin silenced the expression of the SOX2 gene and enhanced the apoptosis of ovarian CSCs [119]. Chung et al. [118] observed a significant inhibition of the viability and proliferation of cancer stem-like cells (OVCAR3 and OVCAR3 CD44+ CD117+ cells) via paclitaxel combined with salinomycin in a dose-dependent manner. The overexpression of SOX2, which is a CSC-associated gene, stimulated cell proliferation, migration, resistance to cisplatin treatment, and the tumorigenicity of ovarian cancer cells [120]. Mi et al. [121] found that salinomycin exerted cytotoxic effects in CD133+ OC cells and decreased the CSC percentage in OC cells. Since salinomycin exhibits poor water solubility, salinomycin-loaded antibody-conjugated nanoparticles have been developed to enable its delivery to CSCs. 

Also, calcium ion channel blockers have been tested as antitumour agents due to the important role of calcium in differentiation, growth, proliferation, and apoptosis, as well as in tumour cells [122]. The importance of calcium channels in tumorigenesis and tumour progression shows the possibility of targeting calcium channels during tumorigenesis. Non-voltage-activated calcium channels, voltage-gated calcium channels, and intracellular calcium channels (including the IP3R and Orai families) have been suggested to be promising in treating cancers. Lee et al. [123] demonstrated that four calcium channel blockers promoted the apoptosis of OCSCs, as they inhibited AKT and ERK signalling. Moreover, calcium channel blockers combined with cisplatin increased drug sensitivity in a CSC-enriched epithelial OC population. In another study, the combination of calcium channel blockers with poziotnib (growth factor receptor inhibitor) resulted in a limited expression of stem cell markers, particularly CD133, NANOG, and KLF4, and the inhibition of STAT5, AKT, and ERK phosphorylation, which translated into a decreased self-renewal ability of OCSCs [124].

Another study demonstrated that the inhibition of Notch signalling with the use of a γ-secretase inhibitor strongly inhibited the self-renewal and proliferation of ovarian CSCs. Moreover, it considerably downregulated the expression of CSC-specific surface markers and decreased the protein levels and mRNA expression of Oct4 and Sox2 in ovarian CSCs [125]. Since the Notch signalling pathway is crucial for ovary CSC maintenance, proliferation, differentiation, and tumour resistance to platinum, other compounds that inhibit Notch have also been studied. Withaferin A (WFA) was demonstrated to suppress tumour growth and eliminate CSCs [126]. 

The phosphoinositide 3-kinase (PI3K)/AKT/mammalian target of rapamycin (mTOR) pathway appears to be an attractive target for anticancer therapy due to the fact that, in majority of cancers, it is associated with aggressive phenotypes, chemoresistance, and poor prognosis [127]. N-t-Boc-hexylenediamine (isoflavone Daidzein) was suggested to trigger apoptosis in OCSCs via the degradation of AKT and the inhibition of the mTOR pathway [128]. Even though the PI3K/AKT/mTOR pathway is an attractive therapeutic target, many drugs have not reached late-phase clinical studies. Clostridium perfringens enterotoxin (CPE) was also suggested to have therapeutical potential in ovarian CSCs. This enterotoxin was found to show high affinity towards the tight junction protein claudin-4. Animal studies demonstrated that injections of CPE were beneficial in chemotherapy-resistant CD44+ OCSC tumours [129]. 

## 6. Ovarian Cancer Stem Cells Markers

Various studies have enabled the identification of markers indicating the presence of ovarian CSCs, including ALDH1, CD24, CD117 (c-kit), CD44+ (hyaluronic acid receptor), and CD133 (prominin-1) [130]. Some markers expressed on the surface of CSCs correlate with clinical features and can potentially be used for the diagnosis of, and the prediction of prognosis in, ovarian cancer [131]. The presence of these specific markers can be used to identify CSCs in ovarian cancer [49]. 

### 6.1. CD133 (Prominin-1)

CD133, a pentaspan transmembrane glycoprotein, is a well-known marker of CSCs and has been identified in various cancers [49]. Although the biological function of CD133 remains unknown, it has been suggested to participate in primitive cell differentiation and epithelial–mesenchymal interactions, autophagy, and, thus, in the malignant progression of ovarian cancer [132]. Moreover, it was suggested to regulate adhesion to sites of metastasis [133]. The expression of *CD133* in ovarian cancer cells is subject to epigenetic regulation via methylation [134]. Ferradina et al. [135] first demonstrated a higher abundance of CD133-1 and CD133-2 epitopes in ovarian tumours than in benign tumours and normal ovarian tissues. Moreover, they revealed the increased clonogenic and proliferative potential of CD133-positive ovarian cancer cells compared with CD133-negative cells [135]. In their study, *CD133* expression correlated with the presence of high-grade serous carcinoma, increased ascites, advanced-stage disease, and a lack of response to chemotherapy [135]. Additionally, a meta-analysis comprising eight studies and 1051 patients with ovarian cancer demonstrated a positive correlation between CD133 levels and tumour stage (odds ratio (OR) = 0.26, 95% CI 0.12–0.58, *p* = 0.001, random effect), as well as an association of *CD133* overexpression with reduced 2-year overall survival (OR = 1.67, 95% CI 1.06–2.63, *p* = 0.03, fixed effect), but not with histological type (OR = 1.10, 95% CI 0.82–1.47, *p* = 0.54, fixed effect) or response to treatment (OR = 0.84, 95% CI 0.61–1.16, *p* = 0.29, fixed effect) [136]. Ruscito et al. [137] observed that *CD133* expression was representative of FIGO stage III/IV patients and was associated with a reduced progression-free interval and overall survival. The observed poor overall survival rate in patients with tumours expressing CD133 suggests the prognostic importance of this marker in ovarian cancer. Another large meta-analysis demonstrated that *CD133* expression correlated with FIGO stage (OR = 3.410, 95% CI 2.196–5.294, *p* < 0.001) and differentiation grade (OR = 2.672, 95% CI 1.354–5.272, *p* = 0.005) [138]. An examination of 45 matched primary and recurrent tumours collected from patients with high-grade ovarian adenocarcinomas demonstrated significantly increased *CD133* expression in recurrent platinum-resistant samples compared with primary tumours [139]. The overexpression of genes encoding members of the TGF-β superfamily, Hedgehog, Notch, and Wnt was demonstrated in CD133-positive recurrent cancer samples [139]. Additionally, a retrospective study aiming to identify predictors of metastases in epithelial ovarian cancer found a positive correlation between *CD133* expression in the primary tumour and platinum resistance, an increased risk of metastases to the central nervous system, and a less favourable prognosis [140]. Patients lacking CD133 clusters, in whom multimodal therapy comprising stereotactic radiosurgery was used, showed improved outcomes of metastatic lesions. A multivariate analysis in 400 ovarian carcinoma samples identified CD133 as an independent predictor of reduced disease-free survival [141]. Moreover, in contrast to ALDH1 and CD44, CD133 was suggested to be a marker of recurrent ovarian cancer. The simultaneous expression of *CD133* and *ALDH1* was suggested to characterise CSCs in ovarian cancer. The expression of both *ALDH1* and *CD133* depends on the appearance of selection pressures, including starvation, in vivo passage, and sphere culture [142]. One study reported that the tumorigenicity of SKOV3 cells expressing ALDH and CD133 is 100 times higher than that of cells expressing ALDH but not CD133, translating into decreased disease-free and overall survival [143]. 

CD133-positive cells isolated from primary ovarian cancers and ovarian cancer cell lines expressed endothelin receptor-A (ETRA), which is involved in cell migration, metastasis, and proliferation [144]. The blockade of ETRA (with the ETRA/ETRB inhibitor macitentan) reduced the percentage of CSCs induced by chemotherapy and lowered the ability of these cells to form spheres [144]. Another study found the upregulation of the chemokine CCL5 and its receptors (CCR1, CCR3, and CCR5) in CD133-positive CSCs, which translated into an increased invasive capacity of the ovarian tumour via the activation of nuclear factor κB (NF-κB) and an increased expression of metalloproteinase-9 (*MMP9*) [145]. Moreover, the inhibition of these receptors resulted in diminished cell aggressiveness. Interleukin-17 produced by the tumour microenvironment stimulated self-renewal in CD133-positive CSCs [146]. Another study demonstrated that the overexpression of *miR-200a* in CD133-1-positive cells was associated with reduced migration and invasion of CSCs by targeting the E-cadherin repressor ZEB2 [147]. Ponnusamy et al. [148] demonstrated that an increased expression of *MUC4* (epithelial cell mucin), a known antigen, was associated with a greater (0.1%) side population of cells sorted using flow cytometry due to their ability to efflux Hoechst 33342 dye and more CD133-positive CSCs. Based on the presented results of various studies, the prospect of ovarian cancer therapy appears promising by directly targeting cancer stem cells (CSCs), which significantly contribute to drug-resistant tumour relapse, using an anti-CD133-targeted toxin.

### 6.2. CD44

CD44 is a glycoprotein expressed on the outer surface of many mammalian cells, including epithelial cells, endothelial cells, leukocytes, and fibroblasts [149]. In humans, the gene encoding CD44 is located on chromosome 11. Approximately 20 isoforms of CD44, formed as a result of alternative splicing and post-transcriptional regulation, have been identified [150]. CD44 is involved in specific cell–cell and cell–extracellular matrix interactions. Its expression has been linked to cancer, vascular disease, interstitial lung disease, infections, arthritis, and other diseases [151,152,153,154]. Many studies have suggested that CD44 is a marker of CSCs in various organs (ovary, breast, colon, prostate, head, etc.) [155,156]. It is involved in cell survival, proliferation, adhesion, and motility [157]. The structure of CD44 comprises four domains: a conserved extracellular hyaluronan-binding domain, an intracellular cytoskeletal/signalling domain, a transmembrane sequence, and variably spliced regions [150]. Some studies suggest that the stem cell niche is enriched in the extracellular matrix glycosaminoglycan hyaluronan, which creates a favourable microenvironment for stem cell self-renewal and maintenance [158]. Hyaluronan can bind to the extracellular domain of CD44, thus affecting its function in CSCs. The interaction of hyaluronan with CD44 triggers the recruitment of various signalling molecules and the activation of the phosphatidylinoside 3-kinase (PI3K) signalling pathway, which activates cellular mechanisms involved in cell survival and invasion, as well as other functions [159]. The interaction of CD44 with hyaluronic acid in ovarian cancer also results in the activation of NANOG–STAT3 [160]. The interaction of NANOG with STAT3 is associated with an increased expression of multidrug resistance protein 1 (*MDR1*) and an enhanced efflux of chemotherapeutic drugs, leading to chemoresistance [14]. Increased hyaluronan production stimulates the acquisition of CSC signatures via EMT [161]. The accumulation of hyaluronan within the tumour stroma was demonstrated to correlate with poor prognosis and reduced overall and disease-free survival in patients with epithelial ovarian cancer [162]. Apart from hyaluronan, the extracellular domain of CD44 can also bind collagen, osteopontin, fibronectin, and laminin [163]. 

The clinical significance and prognostic value of CD44 as a CSC surface marker in patients with ovarian cancer remain controversial. CSCs expressing CD44 were demonstrated to be resistant to chemotherapy and associated with poor survival in ovarian cancer in many studies [164,165,166]. Of 96 patients with stage IIIB-IVA primary serous epithelial ovarian cancer, 49% showed variable *CD44* expression [156]. The presence of CD44-positive tumours was associated with significantly reduced disease-free (*p* ≤ 0.001) and overall survival (*p* ≤ 0.001). The hazard ratio for the death of patients with CD44-positive cancer was 6.8 (95% CI 2.4–19.2, *p* ≤ 0.001), and carboplatin-resistant or carboplatin-refractory disease was an independent predictor of mortality. Based on these observations, the authors concluded that the expression of *CD44* contributed to the development of carboplatin resistance in patients with advanced serous epithelial ovarian cancer, which translated into worse prognoses [156]. 

A meta-analysis based on 18 studies involving more than 2000 patients with ovarian cancer demonstrated that the expression of *CD44* correlated with a high TMN (Classification of Malignant Tumour) stage and a poor 5-year overall survival. However, the authors did not observe an association between CD44 and tumour grade, lymphatic metastasis, response to chemotherapy, or disease-free survival [167]. Another meta-analysis showed the negative impact of *CD44* overexpression on overall and disease-free survival [138]. Moreover, *CD44* expression was observed in patients with chemotherapy-resistant ovarian cancer. Zhou et al. [164] observed a relationship between high *CD44* expression and histological grade, a more advanced FIGO stage, and poorer disease-free survival. The expression of *CD44* was also found to be increased in patients with chemotherapy-resistant epithelial ovarian cancer [168]. An analysis of 138 ovarian tissue specimens revealed that the co-expression of *CD44* and myeloid differentiation factor 88 (*MYD88*) in patients with epithelial ovarian carcinoma correlated with tumour progression, metastasis, and recurrence [165]. According to the authors, *CD44*/*MYD88* co-expression was an independent factor for poor disease-free and overall survival. Cells positive for CD44 and MyD88 in epithelial ovarian cancer demonstrated increased cytokine/chemokine production, an enhanced formation of spheroids, increased repair capacity, and chemoresistance [169]. 

Additionally, CD44 variant 6 (CD44v6) is highly expressed in ovarian cancers, implying that it plays an important role in the development and progression of this cancer [170]. CD44v6-positive cancers were found to be resistant to chemotherapy and to have greater metastatic potential and an increased risk of shortened overall survival [168]. Motohara et al. [171] linked the expression of *CD44v6* to the risk of distant metastatic recurrence and reduced overall survival in patients with ovarian cancer. In vitro and in vivo tests showed that small interfering RNA targeting CD44 in combination with paclitaxel delivered via a cancer-targeted delivery system resulted in the induction of cell death and reduced the tumour [172]. Such therapy effectively suppressed CD44 mRNA and protein levels, thus reducing the population of CD44-positive cancer stem-like cells, and it had no serious side effects.

An analysis of CD44 isoforms in 254 tumour samples obtained from The Cancer Genome Atlas RNAseqV2 demonstrated that patients with a high expression of the *CD44v8-10* isoform have longer survival [173]. The authors found a relationship between the presence of CD44v8-10 on the surface of primary tumour cells and an epithelial phenotype, as well as superior prognosis. By contrast, the expression of the soluble extracellular domain of *CD44* in ascitic fluid was associated with a worse prognosis (*p* < 0.05) [173]. Therefore, it has been suggested that the presence of transmembrane CD44v8-10 on the surface of primary tumour cells may be considered a marker of a highly epithelial tumour with a better prognosis. 

The utility of the combination of CD44 and CD117 as a marker of epithelial ovarian stem-like cells has also been analysed. Zhang et al. [174] demonstrated the ability of cells expressing CD44/CD117 to recapitulate the original tumour in vivo. The population of CD44/CD117-positive cells was found to be increased by the presence of tissue transglutaminase (TG2) induced via the TGF-β-mediated pathway [175]. Moreover, after exposure to low doses of cisplatin, SKOV3 cells showed an enhanced expression of *CD44*, *CD117*, and *ALDH1,* as well as the features of EMT. Moreover, these cells had the ability to form spheres; displayed higher motility; and showed multidrug resistance, the upregulation of cytochrome C, and increased mitochondrial mass [176]. By contrast, a reduction in *CD44* and *CD117* expression on SKOV3 stem cells and the loss of stem-like properties were associated with the overexpression of *miR-200c*. A higher expression of *miR-200c* also resulted in the diminished expression of the genes encoding ZEB-1 and vimentin, as well as the upregulation of E-cadherin [177]. Meng et al. [178] demonstrated that the CD44-positive CD24-negative phenotype of ovarian cancer correlated with a higher recurrence rate and decreased progression-free survival. CD44, being specifically localised in cancer cells rather than borderline serous tumours, emerges as a suitable therapeutic target marker with a focus on cancer stem cells (CSCs) in high-grade serous ovarian carcinoma (HGSC).

### 6.3. ALDH1

ALDH1 has been suggested to be a prognostic marker for various cancers, including ovarian, breast, pancreatic, lung, and colon cancers [179]. It regulates various pathways involved in carcinogenesis and stem cell signalling. This enzyme was suggested to be regulated by potentially oncogenic pathways, including Wnt/β-catenin and MUC1-C/ERK, as well as by chemotherapeutic retinoids [179]. *ALDH* expression and activity are also subjected to regulation through NF-κB signalling via an alternative RelB-dependent pathway [180]. ALDH catalyses the oxidisation of aldehydes that participate in signalling mechanisms or induce cellular or DNA damage, and they also participate in protection from ROS [181,182]. Through these mechanisms, ALDH may exert an impact on cellular differentiation, stemness, tumorigenesis, resistance to therapy, and tumour recurrence [183]. The impact of *ALDH1A1* expression on treatment resistance was initially observed in taxane- and platinum-resistant ovarian cancer cells [184]. In this study, high *ALDH1A1* expression also inversely correlated with survival. The ovarian cells that were positive for ALDH and CD133 were able to initiate tumour development from as few as 11 cells [143].

Several studies have reported an association between high *ALDH1* expression and poor prognosis in patients with ovarian cancer [185,186,187]. Clark et al. [179] provided evidence for the regulation of *ALDH1A1* expression by the Wnt signalling pathway. Moreover, they demonstrated that ALDH activity stimulated the CSC phenotype and resistance to radiation therapy via boosted DNA repair processes and reduced ROS. A large meta-analysis of 18 studies and more than 2500 patients showed that a higher *ALDH1* expression was associated with poor overall survival but had no significant impact on disease-free survival. Increased ALDH1 levels occurred most frequently in patients with unfavourable clinicopathological characteristics. Furthermore, the overexpression of *ALDH1* correlated with a more advanced FIGO stage, lymph node metastasis, and distant metastasis [188]. Similarly, a meta-analysis of 52 studies found a relationship between *ALDH1* expression and FIGO stage (OR = 1.872, 95% CI 1.14–3.076, *p* = 0.013) and lymph invasion (OR = 2.78, 95% CI 1.08–7.152, *p* = 0.034) [138]. ALDH1 has been suggested to be at least partly responsible for the protection of CSCs against chemotherapy [189]. Therefore, Uddin et al. [189] conducted a study to assess whether high ALDH1 CSC positivity rates might be indicative of poor treatment response to cisplatin. They used the cisplatin-sensitive ovarian cancer cell line A2780, the resistant population A2780-Cp, and a supra-resistant population (SKOV3-Cp) from a naturally cisplatin-resistant cell line SKOV3. Both resistant/supra-resistant cell lines displayed a considerably greater self-renewal capability than their parental counterparts and more frequently expressed the ALDH1 marker. Moreover, the silencing of ALDH1 using siRNA reduced the NEK-2 expression level and protein concentration, significantly decreasing the population of stem cells, which are cells sensitised to cisplatin [189]. Similar results were obtained in a study of formalin-fixed, paraffin-embedded tissues collected from 347 ovarian cancers analysed via microarray [190]. In this study, Roy et al. [190] found that ALDH1A1-positive tumours were three times more likely than ALDH1A1-negative tumours to show platinum refractoriness (17% vs. 6; *p* = 0.04). However, they did not observe any significant correlation between ALDH1A1 status and progression-free or overall survival.

However, Chang et al. [191] observed an association between *ALDH1* expression and favourable prognosis in ovarian cancer. In their study, *ALDH1* expression was analysed in 442 primary ovarian carcinomas via tissue microarray and immunostaining. A high *ALDH1* expression was significantly more frequent in endometrioid adenocarcinoma (*p* < 0.0001), in early-stage disease (*p* = 0.006), in samples from patients with low serum levels of CA125 (*p* = 0.02), and in patients with a complete response to chemotherapy (*p* < 0.05). Moreover, a high number of cells expressing *ALDH1* correlated with longer overall survival (*p* = 0.01) and disease-free survival (*p* = 0.006). Similarly, in a study of 248 paraffin-embedded, formalin-fixed ovarian carcinoma tissues obtained from patients included in a long-term follow-up study, a relationship was identified between high *ALDH1* expression and less advanced histological subtypes, early FIGO stage, and better survival probability (*p* < 0.05) [192]. However, no association was observed between *ALDH1* expression in stromal cells and clinicopathological factors (*p* > 0.05). 

These discrepancies between studies may be associated with the use of different study designs; the use of a manual scoring system, which may have induced a level of subjectivity in some studies; and the application of different cut-off points in the analyses; they may also be due to the fact that, in some studies, samples were assessed retrospectively (using archived specimens). Nevertheless, targeting ALDH may present a potential approach to overcome therapeutic platinum resistance.

### 6.4. CD24

CD24 is a mucin-type adhesion molecule localised in lipid rafts via its glycosylphosphatidylinositol anchor [49]. Human CD24 is located on chromosome 6q21 [193]. Various CD24 isoforms differing in molecular mass and cell-type-dependent glycosylation pattern have been isolated from different tissues and cells [194]. Hematopoietic cells (e.g., B cells and T cells) and non-hematopoietic cells (neurons, epithelial stem cells, and epithelial cells) can express CD24 [194]. The chief role of CD24-positive cells remains mostly unknown; however, CD24 may be involved in some immune-regulatory functions [195]. The diffuse cytoplasmic accumulation of CD24 has been reported in cancer cells [196]. Most published studies show an association between *CD24* expression and advanced disease stage and poor prognosis. The utility of CD24 as an independent prognostic marker of survival has been suggested in patients with ovarian cancer [197]. In patients who underwent primary surgery, a higher CD24 expression was considerably associated with poor survival [197]. Nakamura et al. [198] observed CD24 expression in 70.1% of primary ovarian carcinoma tissues. Their study revealed that *CD24* expression correlated with a more advanced FIGO stage and peritoneal and lymph node metastases, and independently predicted survival. CD24 stimulates EMT, which plays a crucial role in colony formation, cell invasion, and cisplatin resistance mediated by the activation of the PI3K/Akt, NF-κB, and ERK signalling pathways [198]. Ovarian cancer cells expressing CD24 were found to display greater potential for spreading (metastasis) to intra-abdominal organs in in vivo models [198]. Due to the stimulation of pathways triggering EMT and cell growth-related intracellular signals by CD24, Nakamura et al. [198] suggested that this molecule plays an important role in metastatic progression and could be a promising therapeutic target in advanced ovarian cancer. In another study, cytoplasmic CD24 was associated with poor survival in ovarian cancer; however, membranous CD24 did not appear to impact patients’ survival [199]. Soltész et al. [200] demonstrated a significant difference in ovarian tissue *CD24* expression between patients with serous ovarian cancer and healthy controls (44.97 ± 68.06 vs. 0.16 ± 0.32, *p* < 0.01). Moreover, the observed expression level correlated with FIGO grading. Finally, Soltész et al. [200] identified proteins that interacted with CD24: LYN, SELP, FGR, and NPM1. Gao et al. [201] found that CD24-positive cells in vitro demonstrated stem-cell-like characteristics of residual quiescence, had the capacity for self-renewal and differentiation, and were more chemoresistant than CD24-negative cells. Moreover, CD24-positive cells expressed greater mRNA levels of stemness genes, such as those encoding Nestin, Bmi-1, β-catenin, Notch1, Notch4, Oct3/4, and Oct4 [201]. The products of these genes participate in the modulation of various stem cell functions. In one experiment, an injection of CD24-positive cells triggered the formation of tumour xenografts in nude mice, while an injection of the same amount of CD24-negative cells failed to trigger this effect [201]. CD24-positive cells isolated from an animal model of ovarian cancer were able to trigger tumorigenesis via JAK2–STAT3 signalling pathways [202].

### 6.5. CD117

The proto-oncogene CD117 (c-kit) is located on chromosome 4 (4q12) and encodes a type 3 transmembrane tyrosine kinase receptor [203]. CD117 was initially identified as the cellular homologue of the feline sarcoma viral oncogene v-kit [203]. Following activation by stem cell factor, CD117 stimulates various signalling pathways [203]. CD117 is involved in the maintenance of cell functions, including cell metabolism, growth and proliferation, survival, differentiation, apoptosis, and migration [203]. The development of CD117’s oncogenic potential is associated with overactivation resulting in the upregulation of the aforementioned pathways or ligand-independent constitutive gain-of-function mutations [203]. Such mutations have been demonstrated in many malignancies [204]. CD117-positive ovarian tumour cells exhibit self-renewal properties and chemoresistance. CD117 signalling has been suggested to be necessary for the maintenance of cell plasticity [203]. The activation of CD117 in cancer results in the activation of downstream signalling pathways that promote stemness or a stem-like phenotype, including RAS/ERK, PI3K, SRC, JAK/STAT, Wnt, and Notch [203]. Indeed, *CD117*-expressing cells isolated from ovarian cancer showed differentiation, self-renewal potential, and stemness [205]. The overexpression of this receptor triggers chemoresistance to cisplatin/paclitaxel via the activation of Wnt/β-catenin-ABCG2 signalling [78]. A meta-analysis conducted by Yang et al. [206] comprising seven studies and 1247 patients with epithelial ovarian cancer found that *CD117* expression significantly correlated with age, FIGO stage, histological type, and tumour differentiation grade. Moreover, patients with a high expression of this receptor had significantly poorer overall survival (hazard ratio (HR) = 1.39, 95% CI 1.03–1.90) than did those with low *CD117* expression. However, the authors did not observe a correlation between *CD117* expression and disease-free survival (HR = 1.31, 95% CI 0.79–2.17) [206]. A subgroup analysis demonstrated that CD117 could play a role as a prognostic factor in European patients and younger patients (<60 years). In another study, a xenograft model was developed by injecting cells from 14 samples of human ovarian serous adenocarcinoma tissue or ascites into mice to assess tumorigenic potential [205]. The obtained CD117-positive lineage-negative phenotype, which comprised less than 2% of xenograft tumour cells, was found to have a 100 times higher tumorigenic potential than CD117-negative lineage-negative cells and resulted in the regeneration of original cancer heterogeneity in a mouse model. Moreover, *CD117* expression was associated with resistance to conventional chemotherapy (*p* = 0.027) in patients [205]. 

The direct targeting of CSCs, which significantly contribute to drug-resistant tumour relapse, with an anti-CD117-targeted toxin exhibits potential as a promising approach for ovarian cancer therapy.

### 6.6. Other Biomarkers Related to CSCs

Leucine-rich repeat-containing G-protein-coupled receptor 5 (LGR5, GPR49, HG38, or FEX) is a glycoprotein hormone receptor involved in the development of malignant tumours [207]. The overexpression of this receptor has been observed in ovarian cancer and associated with cell proliferation, metastasis, and epithelial–mesenchymal transition via the Notch1 signalling pathway [207,208]. In turn, the downregulation of LGR5 was found to suppress the proliferation and hamper the tumorigenicity of ovarian cells in vitro. Moreover, it was associated with the upregulation of the epithelial marker E-cadherin and reduced the levels of mesenchymal markers, such as N-cadherin and vimentin [207]. Additionally, LGR5 knockdown resulted in the downregulation of an EMT-related transcription factor, Snail. The results of experiments with tumour xenografts in nude mice revealed that LGR5 showed strong tumorigenic properties. The role of LGR5 in the stimulation of tumour invasion and metastasis has also been demonstrated in other cancer cells [207].

A type I transmembrane glycoprotein, the epithelial cell adhesion molecule (EpCAM), regulates intercellular adhesion and has been suggested to be present on ovarian cancer cells [209]. EpCAM-positive OC cells exhibit higher tumour-initiating potential than EpCAM-negative cells. The expression of EpCAM was demonstrated to be increased in the tumours of chemo-resistant patients and associated with unfavourable outcomes [210].

The results of various studies have suggested that the expressions of stem cell markers, such as NANOG, SOX2, forkhead-box protein M1 (FOXM1), and OCT4, in ovarian cancer mirror their tumorigenic potential, as well as resistance to paclitaxel, cisplatin, methotrexate, and adriamycin cells [211,212]. Under physiological conditions, NANOG is responsible for the self-renewal and the pluripotency of embryonic stem cells (ESCs) [213]. In ovarian cancer, it regulates self-renewal but also EMT and chemo-resistance via the STAT3 signalling pathway [214]. Yun et al. [214] demonstrated a correlation between the expression of NANOG in OCSCs and a high grade and resistance to standard chemotherapy. Sex-determining region Y-box 2 (SOX2) plays a similar role to NANOG in the physiological state [215]. However, its overexpression was found to trigger cell stemness due resistance to apoptosis related to the inhibition of the PI3K/AKT pathway [216]. Octamer-binding transcription factor-4 (OCT4) participates in embryonic development and cellular pluripotency as a stabiliser of the higher-order chromatin structure in the NANOG locus [211,217]. The expression of OCT4 in the cytoplasm is involved in the regulation of EMT transformation, and its higher levels correlate with tumour initiation and chemo-resistance and predict adverse clinical outcomes. The overexpression of the aforementioned three markers was reported in tumour tissues, as well as in ascites and spheres built from OCSCs [211]. 

FOXM1 belonging to the FOX family of transcription factors is another marker whose overexpression was demonstrated in ovarian CSCs [218]. This factor is an important regulator of the cell cycle and progression, as well as the maintenance of genomic stability. The exposition of ovarian CSCs to lysophosphatidic acid in ascites fluid in OC patients was associated with an increased expression of the FOXM1 protein. The presence of higher FOXM1 levels was associated with the stimulation of wingless and Int-1 (Wnt)/β-catenin signalling. FOXM1 was suggested to protect ovarian cancer cells from the cytotoxicity of cisplatin; thus, its high expression translates into chemo-resistance [219].

CD166 (activated leukocyte cell adhesion molecule, ALCAM), is a transmembrane glycoprotein belonging to the immunoglobulin superfamily that was demonstrated to be overexpressed in various cancers [220]. CD166 stimulates the expression and activation of RAC-alpha serine/threonine-protein kinase (AKT) and the yes-associated protein (YAP). Kim et al. [220] demonstrated that CD166 exhibited CSC-like properties in primary epithelial ovarian cancer cells and promoted the expression of CSC markers, including OCT4, SOX2, and ALDH1A1, as well as ABC transporters.

Autotaxin (ATX) is a tumour cell motility-stimulating factor that promotes the cell motility and cell proliferation of cancer cell lines via the production of lysophosphatidic acid [221]. The expression of ATX has been observed in various tissues, including the ovary. The overexpression of ATX has been found to stimulate tumour motility and invasiveness; increase metastatic potential, resistance to chemotherapy, and radiation-induced cell death; and translate into poor outcomes in cancer patients [222]. ATX was demonstrated to be responsible for maintaining ovarian CSCs through the LPA–LPAR axis since LPA production in sphere-forming cells increased migration; sphere formation; and the expression of CSC markers, including OCT4, SOX2, Kruppel-like factor 4 (KLF4), and ALDH1 [223]. 

High LIN28-expressing ovarian cancer cells that secrete exosomes have been implicated in inducing EMT-related gene expression, invasion, and migration when internalised by non-metastatic target cells [224]. Another study demonstrated the enrichment of ovarian cancer tissue proteins in exosomes, including the epithelial cell surface antigen (EpCAM), epidermal growth factor receptor (EGFR), fatty acid synthase (FASN), proliferation cell nuclear antigen (PCNA), apolipoprotein E (APOE), tubulin beta-3 chain (TUBB3), claudin 3 (CLDN3), L1 cell adhesion molecule (L1CAM), and Erb-B2 Receptor Tyrosine Kinase 2 (ERBB2), which could serve as potential diagnostic markers and therapeutic targets for ovarian cancer. EpCAM, a well-studied biomarker, may have limitations in early-stage ovarian cancer detection, considering its potential cleavage from exosomes via serum metalloproteinase [225]. Hsu et al. observed that EpCAM-regulated transcription was associated with modified biophysical properties of cells that stimulated EMT in advanced endometrial cancer [226]. CLDN3 was deemed less useful than CLDN4, showing 51% and 98% specificity for the detection of ovarian tumours [26]. Elevated claudin-4 expression has been correlated with a phenotypically aggressive ovarian cancer cell, characterised by chemoresistance, high mobility, and stem-like properties [227]. The early identification of the resistance to platinum-based therapy is crucial for improving ovarian cancer treatment. The results of studies have demonstrated that annexin A3 upregulation could be associated with platinum resistance in ovarian cancer [227]. 

Malignant ascites-derived exosomes are found to contain various cargos, including L1CAM, CD24, ADAM10, Claudin-4, and EMMPRIN, crucial for tumour progression [228]. CSCs exhibit the preferential overexpression of VEGF receptors (VEGFR-1 and VEGFR-2), promoting angiogenesis and tumour progression. Additionally, CSCs express vascular–endothelial cadherin (VE-Cadherin), Notch, MMP-2, MMP-9, and CXCR4, influencing endothelial cell transformation and promoting angiogenesis in the tumour microenvironment (TME) [229]. TAMs are recruited to ovarian cancer cells by factors released by cancer cells, leading to an immunosuppressive TME [230]. Ovarian CSCs contribute to M2-polarised TAMs through the release of cycloxygenase2 and CCL2, maintaining OCSC stemness. Regulatory T cells (Tregs) are recruited by CSCs via the CCL22 and TGF-β pathways, while hypoxia upregulates CCL2, attracting Tregs and downregulating effector T-cell responses [231,232].

## 7. Conclusions and Future Directions

Great efforts have been made to understand the biology and special properties of CSCs. The application of CSC markers in the evaluation of prognosis and treatment resistance is constantly growing. In this review, we focused on the expression of the ovarian CSC markers *CD133*, *CD44*, *CD24*, *CD117*, and *ALDH1*, which show potential prognostic significance. The role of CSCs in ovarian cancer initiation and progression and their impact on resistance to therapy are still the subjects of many studies. However, the heterogeneity and plasticity of CSCs pose challenges for the determination of specific phenotypes and the accurate identification of CSCs, and they are the source of conflicting study results. The identification of CSC phenotypes could enable the development of more effective diagnostic and therapeutic strategies in ovarian cancer. This is especially important since strategies to overcome resistance to conventional treatments and prolong patient survival are highly awaited. Targeting CSC markers remains a challenge since the expression of known surface markers has also been observed on normal stem cells (embryonic and/or adult stem cells) and sometimes on various normal tissue cells [233]. Considering the development of CSC resistance to treatment, it seems reasonable that combined treatments that target CSCs may be more effective, and, therefore, such approaches will be a new direction in the future. The recognition of the importance of CSCs in various cancers has resulted in the development of treatment strategies targeting these cells, including adaptive T cells, dendritic cells, oncolytic viruses, and immunological checkpoint inhibitors [234]. Monoclonal antibodies targeting specific CSC markers also seem to be a promising therapeutic option. Moreover, chimeric antigen receptor (CAR)-T-cell therapy could be used to target CSC markers [235]. However, despite the considerable progress made in the field of CSCs, a lot of information is still lacking. The identification and use of CSC markers may enable the earlier diagnosis of ovarian cancer, a better understanding of resistance mechanisms, and possibly the targeting of CSCs as an effective treatment strategy. There are over 70 trials on Clinicaltrials.gov. focusing on cancer stem cells; however, only 7 refer to ovarian cancers (3 completed, 1 recruiting, 1 suspended, 1 terminated, and 1 with unknown status). There is a need for more large studies and clinical trials evaluating agents targeting ovarian CSCs. Data obtained from them could improve future cancer detection, treatment, and the survival of females with ovarian cancer. 

## Figures and Tables

**Figure 1 cancers-16-00040-f001:**
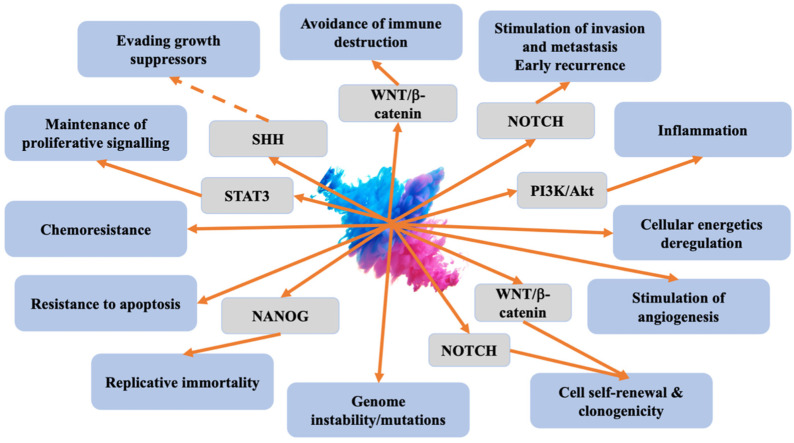
Mechanisms and pathways related to CSCs.

**Table 1 cancers-16-00040-t001:** Summary of clinical trials related to cancer stem cells.

Identifier/Official Title	Phase	Population/Intervention/Treatment	Primary Endpoints	Results
Completed Trials				
NCT02178670 Study of Cancer Stem Cell Vaccine That as a Specific Antigen in Metastatic Adenocarcinoma of the Ovarian	1, 2	40 participants with stage III epithelial ovarian cancer in remission after surgery (hysterectomy and ovariectomy) and the first primary chemotherapy Treatment: biological: CSC-DC	Primary outcome: Safety of immunisation with the cancer stem cell vaccine (number of participants with an adverse event) Secondary outcome: - Immune responses to the immunisations (body measurements, cellular immunity, and humoral immunity) Other outcome: - The dose of CSC vaccine	No results have been published
NCT01579812 A Phase II Evaluation of Metformin, Targeting Cancer Stem Cells for the Prevention of Relapse in Patients with Stage IIC/III/IV Ovarian, Fallopian Tube, and Primary Peritoneal Cancer	2	90 participants with advanced ovarian, fallopian tube, or primary peritoneal cancer	Primary outcome: Recurrence-free survival (time frame: 18 months) Primary endpoints: - ≥2-fold reduction in ALDH+ CD133+ CSCs - >50% relapse-free survival at 18 months Secondary outcome: Overall survival (time frame: up to 3 years)	- Tumours treated with metformin had a 2.4-fold reduction in ALDH+ CD133+ CSCs and higher sensitivity to cisplatin ex vivo - Metformin altered the methylation signature in CA-MSCs, which prevented CA-MSC-driven chemoresistance in vitro - Median PFS was 18.0 months (95% CI 14.0–21.6) with relapse-free survival at 18 months of 59.3% (95% CI 38.6–70.5) - Metformin was well tolerated Conclusions: Translational studies confirmed the impact of metformin on epithelial ovarian cancer CSCs. Metformin therapy was associated with better-than-expected overall survival
Ongoing Trials				
Identifier/official title	Phase	Population/intervention/treatment	Primary endpoints	
NCT03949283 Standard Chemotherapy Versus Cancer Stem Cell Assay Directed Chemotherapy in Recurrent Platinum Resistant Ovarian Cancer	3	150 participants with recurrent platinum-resistant ovarian cancer Diagnostic test: ChemoID assay Drug: standard chemotherapy	Primary outcome: ORR in patients with recurrent epithelial ovarian cancer who had ChemoID-guided treatment versus physician choice control treatment Secondary outcomes: - PFS in patients with recurrent epithelial ovarian cancer who received standard therapy chosen by the physician versus ChemoID drug response assay-directed chemotherapy - Duration of response - CA125 levels - HRQOL	
NCT03632798 Avastin Plus Chemotherapy vs. Avastin Plus Chemotherapy Chosen by Cancer Stem Cell Chemosensitivity Testing in the Management of Patients with Recurrent Platinum-Resistant or -Sensitive Epithelial Ovarian, Fallopian Tube, or Primary Peritoneal Cancer	3	300 participants experiencing first, second, or third recurrence of any stage of epithelial ovarian cancer Diagnostic test: ChemoID assay Drug: chemotherapy	Primary outcome: PFS in patients with recurrent epithelial cancer who received standard treatment with bevacizumab plus chemotherapy chosen by the physician versus bevacizumab plus ChemoID drug response assay-directed chemotherapy Secondary outcome: - Median overall survival - ORR (partial or complete response by RECIST v1.1) - HRQOL	
NCT05576519 Immunohistochemical Expression of Epithelial Cell Adhesion Molecule (EpCAM) in Epithelial Ovarian Carcinoma	*	50 paraffin blocks of epithelial ovarian cancer collected from patients who underwent surgery	Primary outcome: Evaluation of *EPCAM* expression in ovarian carcinoma	
NCT02713386 A Phase I/II Study of Ruxolitinib with Front-Line Neoadjuvant and Post-Surgical Therapy in Patients with Advanced Epithelial Ovarian, Fallopian Tube or Primary Peritoneal Cancer	1,2	147 participants with advanced epithelial ovarian, fallopian tube, or primary peritoneal cancer Drug: carboplatin Drug: paclitaxel Drug: ruxolitinib phosphate Procedure: therapeutic conventional surgery	Primary outcome: - Incidence of haematological (heme) dose-limiting toxicity (phase I) - PFS (phase II) (up to 5 years) Secondary outcomes: - Incidence of adverse events (phase I) (up to 5 years) - Frequency of patients who could not receive surgery within the defined timeframe for reasons other than non-response, disease progression, or medical contraindications (phase I) - Number of patients who discontinued ruxolitinib in the first 3 months of maintenance therapy due to toxicity (phase I) - PFS (phase II) (up to 5 years) - Proportion of patients who had total gross resection (phase II) - Complete pathological response (phase II) (up to 5 years) - Overall survival (phase II) (up to 5 years) Other outcomes: - Change in CSCs observed in tissue (up to 63 days) - Change in serum CRP	

* Retrospective study. Abbreviations: ALDH—aldehyde dehydrogenase; CA-MSCs—carcinoma-associated mesenchymal stem cells; CRP—C-reactive protein; CSCs—cancer stem cells; CSC-DC—CSC-loaded dendritic cells; ORR—objective response rate; PFS—progression-free survival; HRQOL—health-related quality of life.

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
