# Peer review of "The Role of Cancer Stem Cell Markers in Ovarian Cancer"

_cancers, 2023, doi:10.3390/cancers16010040_

Round 1

Reviewer 1 Report

Comments and Suggestions for Authors

Thank you for submitting for review the manuscript on CSC’s and their markers in ovarian cancer.

This is an important and interesting review of this significant element of the fight against ovarian cancer.

My comments:

1. In the abstract, the authors write that ovarian cancer is the third most common cancer in women. That's not true. There is also lung and colon cancer.

2. Instead of "Subtypes", "Stages" would be appropriate (l.51)

3. I think that there are too many citations in the manuscript (almost 180)

4. I am not sure if all citations are quoted correctly, e.g.: [20] l.78; [28.28] l.115

5. Not all citations are presented correctly in references, e.g. [26]

6. I think the whole introduction is unnecessary. It presents generally known knowledge about ovarian cancer.

7. EMT is The epithelial–mesenchymal transition, not transformation. (l.109)

8. In the treatment of ovarian cancer, the first-line drug is carboplatin, not cisplatin, as reported by the authors. This also results from the research they cite (l.135)

9. Is Fig. 1 made by the authors or is it cited? Are there copyrights to reproduction?

10. There is no citation for the use of metformin in ovarian cancer. (l.264)

11. The manuscript contains minor editorial errors (e.g.: l.10, l.43, etc.). This needs improvement.

After corrections and shortening of the description, the manuscript is ready for publication.

Comments on the Quality of English Language

Minor editing of the English language required

Author Response

We would like to thank you for your valuable comments which helped to improve this manuscript. Your suggestion was taken into consideration and appropriate information was provided. We used track changes to facilitate the monitoring of introduced changes. We did our best to meet your expectations and we hope that you will be satisfied with our corrections.

1. In the abstract, the authors write that ovarian cancer is the third most common cancer in women. That's not true. There is also lung and colon cancer.

This information was updated

2. Instead of "Subtypes", "Stages" would be appropriate (l.51)

This part was removed to shorten the introduction

3. I think that there are too many citations in the manuscript (almost 180)

We did our best to decrease the number of citations, however, due to the fact that we were suggested to add many new information the number of referenced articles remained large

4. I am not sure if all citations are quoted correctly, e.g.: [20] l.78; [28.28] l.115

Citations were corrected

5. Not all citations are presented correctly in references, e.g. [26]

Citations were corrected

6. I think the whole introduction is unnecessary. It presents generally known knowledge about ovarian cancer.

The introduction was significantly shortened, however, it was not removed completely. We decided to leave some information to introduce the reader to the topic of the article

7. EMT is The epithelial–mesenchymal transition, not transformation. (l.109)

This word was corrected

8. In the treatment of ovarian cancer, the first-line drug is carboplatin, not cisplatin, as reported by the authors. This also results from the research they cite (l.135)

This information was corrected

9. Is Fig. 1 made by the authors or is it cited? Are there copyrights to reproduction?

The Fig 1 was prepared by authors on the basis of various schemes.

10. There is no citation for the use of metformin in ovarian cancer. (l.264)

The citation was provided

11. The manuscript contains minor editorial errors (e.g.: l.10, l.43, etc.). This needs improvement.

We did our best to correct all editorial errors

Reviewer 2 Report

Comments and Suggestions for Authors

This manuscript is focused on investigating the presence of cancer stem cells in ovarian cancer. Its main objective aims to provide a comprehensive review of the fundamental biology and significance of these cells, along with examining their core markers: CD44, CD133, ALDH1, CD117, and CD24. Despite presenting current information regarding cancer stem cells and the importance of their markers in ovarian cancer, as well as their potential application in therapy and diagnostics, this manuscript lacks distinctive elements that would differentiate it from other similar contemporary works. The presented research doesn't exhibit a high level of novelty and originality. Therefore, it is recommended that the authors revise and expand the relevant sections of the publication. At the very least, they should consider exploring alternative markers to differentiate CSC subpopulations, such as CD34, EpCAM, LGR5, and others. Furthermore, it is advised to thoroughly review the potential for developing CSC-specific therapies and analyzing the results of ongoing clinical trials in greater depth. Implementing these changes will enhance the quality and originality of the work.

Comments on the Quality of English Language

Minor editing of English language required

Author Response

We would like to thank you for your valuable comments which helped to improve this manuscript. Your suggestion was taken into consideration and appropriate information was provided. We used track changes to facilitate the monitoring of introduced changes. We did our best to meet your expectations and we hope that you will be satisfied with our corrections.

1. Despite presenting current information regarding cancer stem cells and the importance of their markers in ovarian cancer, as well as their potential application in therapy and diagnostics, this manuscript lacks distinctive elements that would differentiate it from other similar contemporary works. The presented research doesn't exhibit a high level of novelty and originality.

Thank you for your opinion. We did not perform a meta-analysis or systemic review that is why this article may lack novelty. However, we did our best to summarize all the most important information concerning cancer stem cells and biomarkers. To increase the originality, we added to the manuscript parts concerning alternative markers, the potential of CSC-specific therapies and the results of clinical trials (according to your valuable suggestion)

2. At the very least, they should consider exploring alternative markers to differentiate CSC subpopulations, such as CD34, EpCAM, LGR5, and others.

We provided additional information

3. Furthermore, it is advised to thoroughly review the potential for developing CSC-specific therapies and analyzing the results of ongoing clinical trials in greater depth. Therefore, it is recommended that the authors revise and expand the relevant sections of the publication. Implementing these changes will enhance the quality and originality of the work.

We provided additional information

4. Comments on the Quality of English Language: Minor editing of English language required

The text has been checked and corrected

Round 2

Reviewer 1 Report

Comments and Suggestions for Authors

Accepted. 

Author Response

Thank You for acceptance of the revised manuscript